# Hydrothermal Preparation of TiO$_2$/Graphite Nanosheets Composites and Its Effect on Electrothermal Behavior

**Chunyu Wang** [1,2,*] , **Weiyao Tian** [2] , **Sibo Kang** [3,*] , **Bo Zhong** [2] , **Chunlin Qin** [2,*] **and Hongyang Wang** [2]

1. State Key Laboratory of Advanced Welding and Joining, Harbin Institute of Technology, Weihai 264209, China
2. School of Materials Science and Engineering, Harbin Institute of Technology, Weihai 264209, China
3. Marine Chemical Research Institute Co., Ltd., Qingdao 266071, China
* Correspondence: wcyadam@126.com (C.W.); qingdaokang@126.com (S.K.); apollohit@163.com (C.Q.)

**Abstract:** Nowadays, carbon materials are supposed to replace the resistance wire made of metal alloy to be the next generation of heat-generating materials due to their excellent electrical conductivity and corrosion resistance. In this study, TiO$_2$/graphite nanosheets (GNs) composite was prepared by chemical exfoliation and hydrothermal methods. XRD, FTIR, and Raman spectra confirm TiO$_2$ particles are on the surface of GNs. SEM photographs show TiO$_2$ nanoparticles covering the surface of the GNs uniformly. We used TiO$_2$/GNs and sodium silicate to produce the electrothermal film coated on the glass. As compared to raw GNs, the heating rate and maximum temperature have greatly improved. In order to find the reasons for the improvement, the BET and zeta potential of TiO$_2$/GNs were tested, and we found that the enhancement of the surface area and the dispersion to the composite by TiO$_2$ particles and sodium silicate make the distribution of GNs more uniform.

**Keywords:** TiO$_2$; GNs; hydrothermal; electrothermal behaviors





## 1. Introduction

Electrothermal material is an important area of research for alternative energy as it deals with conductive materials, which convert electricity into heat. Traditional electrothermal materials, such as Ni–Cr (nichrome) or Fe–Cr–Al (Kanthal), because of their heavy weight, rigidity, brittleness, and intolerance to acids or bases, have many restrictions on the application. Therefore, it is necessary to find a material that has the potential to replace them in the future.

Graphene is a two-dimensional carbon material, which has been characterized by excellent electrical properties [1], faster and more efficient heating [2,3], and uniform temperature distribution [4]. More importantly, compared with metal material, graphene is lighter, so it can show the same performance with lighter quality. Moreover, graphene powder disperses easily in an organic solvent, so it can be compounded with other materials to form thin films. Because of the above advantages, graphene is often used as an electrothermal film material.

Despite the above-mentioned merits, graphene as a single component cannot satisfy all demands for many different applications [5]. Until now, different modification and doping strategies have been applied to graphene. Inducing oxygen functional groups at the graphene surface, such as nanofibrillated cellulose (NFC), carboxylic, carbonyl, and hydroxyl groups, can improve the solution dispersibility of graphene [6]. Heteroatom doping can greatly influence the charge-storage capacity of graphene because of the modification of the electronic structure and surface energy [7]. It is generally used in supercapacitors. As in the case of the electrothermal material, simply adding graphene to the polymer coating cannot get the ideal heat material, it is necessary to modify the graphene to improve the performance.

Graphite nanosheets (GNs), which are composed of two-dimension graphite layers, have similar properties to graphite, such as lower density, increased charge carrier mobility,

excellent electrical conductivity, etc. [8]. Besides, compared with the preparation methods of graphene, the preparation of GNs is easier, safer, and friendlier to the environment [9]. Therefore, in this study, GNs is used to prepare electrothermal composite material. Chen developed a novel wearable heater based on flexible, stretchable graphite nanoplates and polyurethane (GNP/PU) nanocomposite films, which can generate heat uniformly under the safe voltage of 5–24 V to achieve rapid heating and cooling rate of 25 °C/min and 13 °C/min, respectively [10].

Titanium dioxide ($TiO_2$) is commonly an inorganic filler used in coating. It has been proved that $TiO_2$ could enhance the thermal stability of the polymer [11], improve the mechanical properties, and endow the materials with high fire resistance [12]. Taking these into account, $TiO_2$-modified graphite nanosheets might be an efficient electrothermal material that has good thermal stability and fire performance. In addition, the stack of the graphite nanosheets is prevented by the tightly anchored nanoparticles, which is beneficial for the uniform dispersion of graphite in a matrix [13–15].

Hence, we modified graphite nanosheets by $TiO_2$ to study the synergistic electrothermal effects on the properties' enhancement of sodium silicate composite film. We prepared GNs by chemical exfoliation of graphite powder, and modified GNs with $TiO_2$ by the hydrothermal method. The preparation method is easy to operate, and the experiment has good results. It is anticipated that loading $TiO_2$ nanoparticles on the graphite nanosheets could prevent the graphite nanosheets from stacking and reduce the resistivity compared with directly adding $TiO_2$. This work attempts to provide a promising electrothermal material so as to enhance the graphite nanosheets' electrothermal behavior.

## 2. Experimental

### 2.1. Materials

Graphite powder (200 mesh) was purchased from Sinopharm Chemical Reagent Co., Ltd (Beijing, China). Titanium oxysulfate ($TiOSO_4$) was obtained from Tianjin Guangfu Fine Chemical Research Institute (Tianjin, China). Potassium permanganate ($KMnO_4$, AP), ethanol, sulfuric acid ($H_2SO_4$, 98%), sodium silicate ($Na_2SiO_3$, 30% aq.), silicone oil, magnesium oxide (MgO), sodium hydrate (NaOH), and antimonous oxide ($Sb_2O_3$) were purchased from Sinopharm Chemical Reagent Co., Ltd (Beijing, China).

### 2.2. Preparation of $TiO_2$/Graphite Nanosheets Composite

Graphite nanosheets (GNs) were prepared by chemical exfoliation of graphite powder. As a typical preparation of GNs, 10 g natural graphite was soaked in 60 mL $H_2SO_4$ and stirred for 10 min, then added 3.5 g $KMnO_4$ and 10 mL acetic acid to the mixture. We kept stirring for 1 h and washed graphite several times with deionized water until the solution was neutral. Next, the product was dried in an oven at 70 °C, then, heated in air atmosphere at 900 °C for 30 s to obtain expanded graphite (EG). The EG was suspended in 70% ethanol. After 40 min mixing by ultrasonic processing under the power of 900 W, GNs were obtained.

The $TiO_2$/graphite nanosheets composite was obtained via the hydrothermal method with modifications. First, we dissolved 0.5 g $TiOSO_4 \cdot 2H_2O$ in 80 mL deionized water, stirring until the solution became completely transparent. Then, we added 0.5 g GNs and 15 mL 1 M HCL to the solution, and it was mixed by ultrasonic processing for 30 min. The resultant solution was transferred into a Teflon-lined autoclave and treated at different temperatures for 12 h. The temperature was set separately at 100, 120 and 140 °C. It was expected that there would be a suitable temperature for the preparation of $TiO_2$ so that that sample could be used in the later steps. The obtained precipitates were washed with deionized water and ethanol and dried at 65 °C.

Heating films were prepared with sodium silicate. Briefly, 2.8 g $TiO_2$/GNs (containing $TiO_2$ 0.8 g) and other additives (Table 1 [16]), except for sodium silicate, were added into deionized water and stirred for 10 min. Then, sodium silicate was added into the mixed suspension and stirred until a homogeneous mixture was obtained. At last, the blend was

cast into a $150 \times 75 \times 0.08$ mm$^3$ film with a 5 mm thick glass plate and dried at room temperature to form flat membranes. Following the above-mentioned methods, films were prepared and subjected to a measurement of electrothermal properties. In the case of the GNs film, the same amount of GNs and TiO$_2$ was added, and the above methods were identical.

**Table 1.** Other additives of electrothermal GNs film.

| Composition | Sodium Silicate | Water | Silicone Oil | MgO | NaOH | Sb$_2$O$_3$ |
|---|---|---|---|---|---|---|
| Contains | 65 g | 60 g | 0.3 g | 0.3 g | 0.1 g | 0.2 g |

*2.3. Characterization*

The phase composition of the product was characterized by X-ray diffraction (XRD, Model DX-2700, Haoyuan, China) with Cu K$\alpha$ ($\lambda$ = 0.15418 nm). The scanning range of this device was from 10° to 90°, and the scanning speed was 4°/min. The Raman spectrometer (Renishaw, Gloucestershire, UK) used He-Ne laser ($\lambda$ = 532 nm) with the power of 20 KW/cm$^2$. The Raman spectrometer was used to distinguish ordered and disordered crystal structures of GNs. Brunauer–Emmett–Teller (BET, Beishide 3H-2000PS1, Beijing, China) measurement could identify the surface area of the sample. The scanning electron microscope (SEM, MERLIN Compact, Zeiss, Oberkochen, Germany) is equipped with EDS, which was used to identify the micromorphology and element composition. The Fourier Transform Infrared (FTIR) spectrometer measurement (Nicolet 380, MA, USA) was used to identify the functional groups of the sample. The zeta potential (Malvern Nano ZS90, Malvern, UK) has the ability to measure the dispersivity of the sample.

### 3. Results and Discussion

*3.1. Characterization of the TiO$_2$/GNs Composite*

In acidic, high-temperature conditions, TiOSO$_4$ is first hydrolyzed to form H$_2$TiO$_3$. The chemical equation is expressed as:

$$TiOSO_4 + 2H_2O \rightarrow H_2TiO_3 + H_2SO_4$$

H$_2$TiO$_3$ is insoluble in water. In order to reduce the nuclear energy, H$_2$TiO$_3$ crystals will be preferred crystallization on the GNs surface. In the presence of strong acid, the dehydration reaction takes place among the hydroxyl groups of H$_2$TiO$_3$ and GNs to form Ti–O–C bonds.

Raman spectroscopy is generally used to characterize the crystal quality of carbonaceous materials. Figure 1 shows the Raman spectra of GNs and the samples in different hydrothermal temperatures. Raman spectra of TiO$_2$/GNs composite have strong Raman characteristic peaks at 148 cm$^{-1}$ (E$_g$(1)), 397 cm$^{-1}$ (B$_{1g}$), 516 cm$^{-1}$ (A$_{1g}$), and 638 cm$^{-1}$ (E$_g$(2)), respectively, which are attributed to a pure anatase phase of TiO$_2$ [17]. As for bare GNs, the Raman spectra are characterized by the D and G bands, arising from one breathing mode of k-point photons of A$_{1g}$ symmetry and the zone center E$_g$(2) mode, respectively [18]. After the TiO$_2$ nanoparticles were loaded on graphite nanosheets, the peaks of TiO$_2$ appeared on the Raman spectra of all the temperature samples, which revealed the successful fabrication of TiO$_2$. We can see that the 140 °C sample had the highest peak of TiO$_2$, it seems that the temperature was more suitable for the preparation of TiO$_2$. Thus, the 140 °C sample was used in the subsequent experiments and tests. For convenience, the temperature was left out in the following text and the composite was named TiO$_2$/GNs. Furthermore, the G band of TiO$_2$/graphite nanosheets composite shows a shift to higher wave numbers while the 2D band shifted to lower wave numbers, compared to that of raw GNs, which is attributed to the effect of doping and the reduced number of sheets, indicating that the re-stack of GNs sheets was controlled. That may be because the TiO$_2$ particles obtained by the hydrothermal method are able to reduce the stack of GNs.

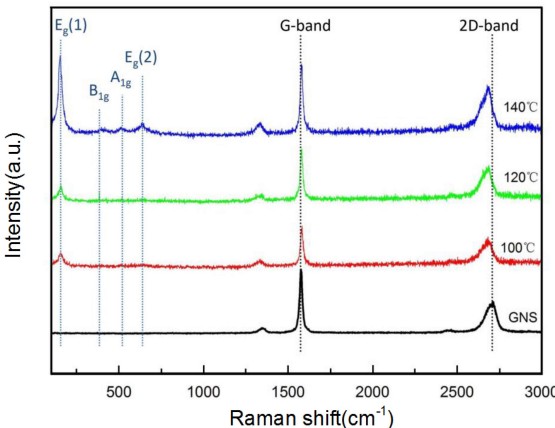

**Figure 1.** Raman spectra of GNs and the TiO$_2$/GNs samples in different hydrothermal temperatures.

Field emission scanning electron microscopy (FESEM) and Energy Dispersive X-ray Spectroscopy (EDS) profiles (see Figure 2) describe a continuous dispersion of TiO$_2$ nanoparticles over graphite nanosheets. For pure GNs, it has a uniform surface morphology revealing a rather smooth surface (Figure 2a). Compared to pure graphite nanosheet, there are some TiO$_2$ nanoparticles existing on the surface (Figure 2b). In enlarged Figure 2b, a thin flocculent film can be observed, and EDS report (Figure 2d) shows the red point area with the elements of titanium and oxygen, which implies that the film may be made up of TiO$_2$. Table 2 shows the relative content of different elements in the EDS report in detail. The film consists of a number of small particles, and we speculate that those large particles may be made up of small particles due to the concentration of the reaction at the defect.

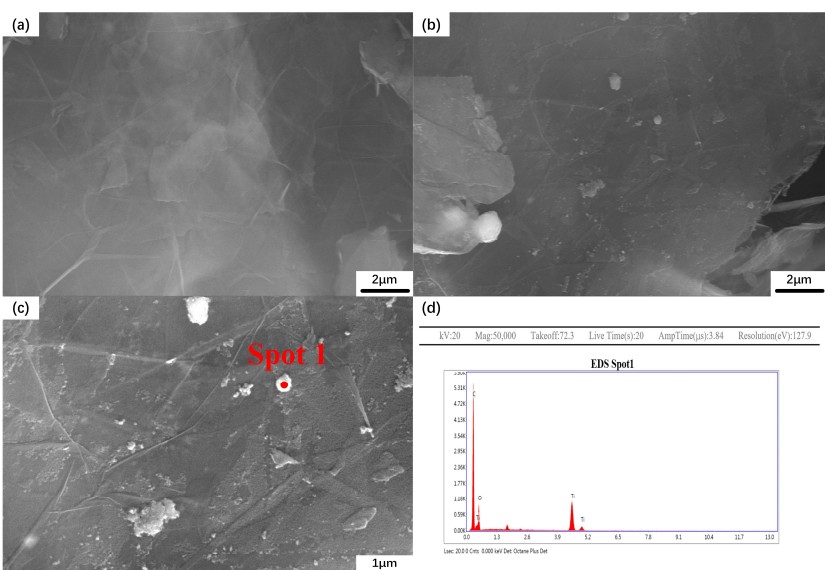

**Figure 2.** SEM images of (**a**) GNs, (**b**) TiO$_2$/GNs composite, (**c**) partial enlarged drawing of (**b**), and (**d**) EDS report at red point in (**c**).

**Table 2.** The relative content of different elements in the EDS report.

| Element | Weight % | Atomic % | Net Int. | Error % | K-Ratio |
|---------|----------|----------|----------|---------|---------|
| C | 63.30 | 74.44 | 3109.42 | 4.74 | 0.4307 |
| O | 25.06 | 22.13 | 526.64 | 10.90 | 0.0316 |
| Ti | 11.63 | 3.43 | 1275.51 | 2.46 | 0.0955 |

X-ray diffraction (XRD) patterns of the as-prepared GNs and $TiO_2$/GNs composite are given in Figure 3. The diffraction peaks of GNs are similar to graphite, which has a strong peak near 26° corresponding to the (002) plane of graphite. The $TiO_2$/GNs composite sample shows $TiO_2$ exists as an anatase phase indicated with peaks at 25.3°, 37.8°, 48.0°, and 55.1°. This result supports the assumption that the white particles in the SEM photograph are $TiO_2$.

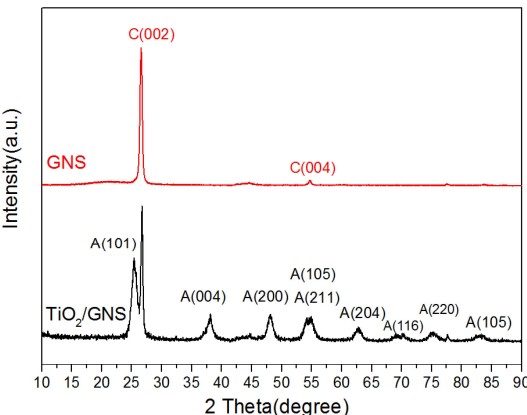

**Figure 3.** XRD patterns of GNs and $TiO_2$/GNs composite.

To determine the specific surface area of the composite nanomaterial, $N_2$ adsorption/desorption measurements and Brunauer–Emmett–Teller (BET) analysis method were used. Nitrogen adsorption/desorption isotherms for GNs and $TiO_2$/GNs composite are shown in Figure 4. The overall shape of the $TiO_2$/GNs indicates a material with mesoporous and macroporous characteristics. It can be calculated by BET analysis method based on Figure 4 that the surface area of the $TiO_2$/GNs composite increased to 58.9 $m^2 \cdot g^{-1}$, which is higher than blank $TiO_2$ and pure GNs. The BET surface area of GNs is calculated to be 20 $m^2 \cdot g^{-1}$. The surface area of $TiO_2$/GNs composite is almost three times as much as GNs. Furthermore, the BET surface area of GNs is higher than raw graphite particles, which is 8.9 $m^2 \cdot g^{-1}$. However, it is still lower than the theoretical specific surface area (2630 $m^2 \cdot g^{-1}$), possibly due to the incomplete exfoliation of graphite oxide and the agglomerations of graphene layers during the reduction process [19].

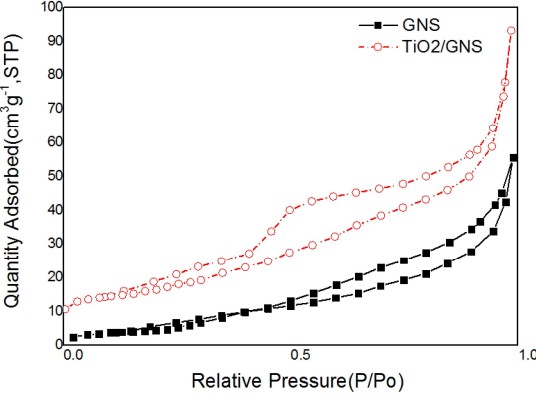

**Figure 4.** Nitrogen adsorption-desorption isotherms for GNs, $TiO_2$/GNs.

FTIR spectra of raw GNs and $TiO_2$/GNs composite are shown in Figure 5. For convenience, the two curves were put together in one figure to be compared. It can be seen that after the thermal reduction of the graphite nanosheets, the oxidation groups on the surface have been completely removed, only a few carbon-hydrogen bonds can be detected. The broader absorption band below 1000 $cm^{-1}$ in $TiO_2$/GNs composite corresponds to the

Ti–O–Ti vibration. A small absorption peak was observed at 1630 cm$^{-1}$, probably due to the small amount of absorbed water on the surface of TiO$_2$.

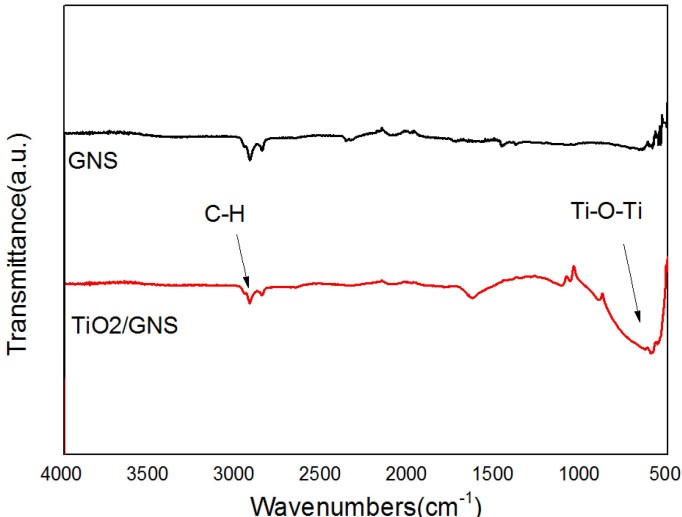

**Figure 5.** FTIR spectra of raw GNs and TiO$_2$/GNs.

Through the zeta potential test, we found that TiO$_2$/GNs composite and raw GNs have different dispersing capacities in sodium silicate solutions. Figure 6 shows the zeta potential of raw GNs and TiO$_2$/GNs composite in 1% sodium silicate solution, which is because the high concentration of sodium silicate causes the conductivity of the solution to increase greatly, resulting in an unmeasurable zeta point. The zeta potential of TiO$_2$/GNs composite has an absolute value of about 30, higher than that of raw GNs about 15. The absolute value of zeta potential is an important indicator of dispersion stability, the larger the number, the better the dispersion [20]. This improvement in dispersing ability is due to the dispersion of TiO$_2$ in sodium silicate. The reason for the wider distribution of GNs is that the amount of TiO$_2$ attached to the graphite nanosheets is different.

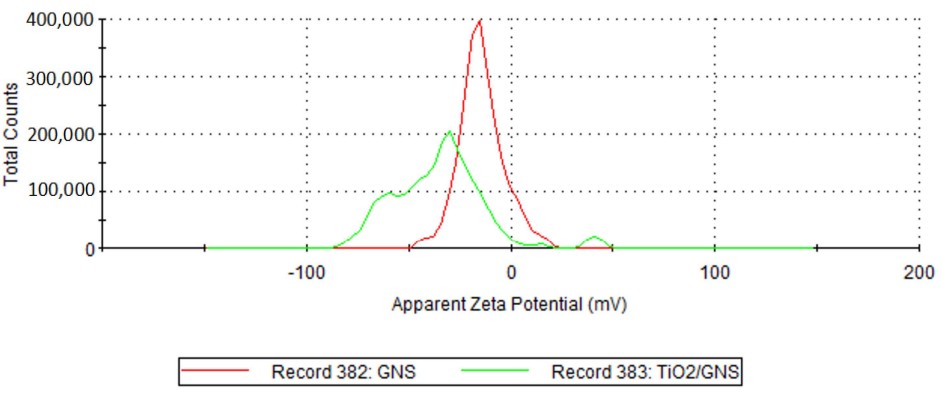

**Figure 6.** Zeta potential of GNs and TiO$_2$/GNs.

### 3.2. Electrothermal Application of TiO$_2$/GNs Composite

Electrical conductivity, heating rate, and maximum temperature rise are the three main parameters of electrothermal properties. Figure 7 displays time-dependent temperature changes of the TiO$_2$/GNs films and raw GNs films at 15 V DC voltage, and the current is about 5 A. The film is heated at room temperature. The curve clearly shows the heating rate and maximum temperature rise of TiO$_2$/GNs film are obviously higher than GNs film. In the maximum temperature rise region (500–600 s), the heat gain by electric power is equal to the heat loss by radiation and convection based on the conservation law of energy, which is in thermal equilibrium. The maximum temperature rise of TiO$_2$/GNs film is about 20 °C

higher than GNs film. Table 3 is a summary of other electrothermal films about the driving voltage, saturated temperature, and heating speed.

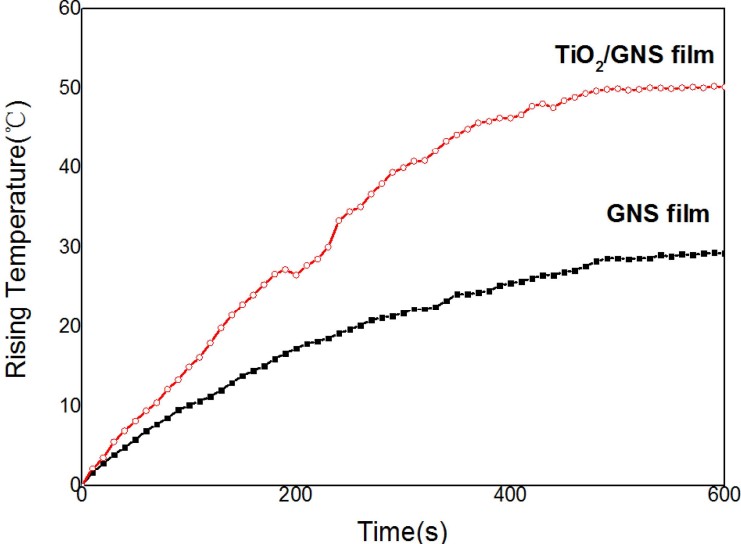

**Figure 7.** Heating curves of TiO$_2$/GNs and GNs films.

**Table 3.** Summary of other electrothermal films.

| Sample | Voltage/Power | Max Temperature (°C) | Speed | Reference |
|---|---|---|---|---|
| Graphene (GE)/UC film | 30 V | 199 | - | [21] |
| GNSs/MWCNTs/CB | 3 V | 175 | 17.5 °C/s | [2] |
| laser-induced graphene/PI flim | - | 270 | 27 °C/s | [3] |
| water-dispersible graphene films | 10 V | 147 | 11.8 °C/s | [8] |
| GNP/PU film | 5–24 V | - | 25 °C/min | [10] |
| GNP films | 3–5 V | - | 25–65 °C/min | [22] |
| graphene sheets/NFC membrane | 2000 W·m$^{-2}$ | 60 | 20 °C/min | [5] |
| RGO/MnFe$_2$O$_4$ paper | 0.35 A | 100.2 | 6.68 °C/s | [23] |
| F-N Co-doped laser reduced GO | 9 V | 365 | 385.33 °C/s | [24] |
| polyurethane/graphene | 22 V | 75 | - | [25] |

The improvement of electrothermal property is due to the decrease of sheet resistance. Using the four-probe method to measure the resistivity of the films at the same amount of GNs added, the resistivity of TiO$_2$/GNs film is about 0.071 Ω·cm, far below the GNs film, which is about 0.135 Ω·cm. The synergistic effects of GNs and TiO$_2$ are proposed to be the possible reason for reducing the resistivity of sodium silicate materials. We speculate that the synergistic effects mainly show in three respects: The TiO$_2$ nanoparticles loaded on GNs prevent the GNs from re-stacking and result in good dispersion. The presence of TiO$_2$ increases the specific surface area, which makes GNs from hydrophobic to hydrophilic. Sodium silicate plays a role in the dispersion of TiO$_2$/GNs. Figure 8 shows that when the dispersion capacity of the GNs is increased, the arrangement between the sheet and sheet tends to be more uniform, and more overlap points cause the resistance to decrease. Figure 9 shows the SEM images of GNs film and TiO$_2$/GNs film. Figure 9a shows that the GNs have a very obvious tendency to agglomerate, and each layer is not well lapped. This structure cannot show the excellent conductivity of GNs. Figure 9b shows that the TiO$_2$/GNs film has a very uniform distribution, which supports that the arrangement of GNs becomes uniform and further makes an impact on the resistance.

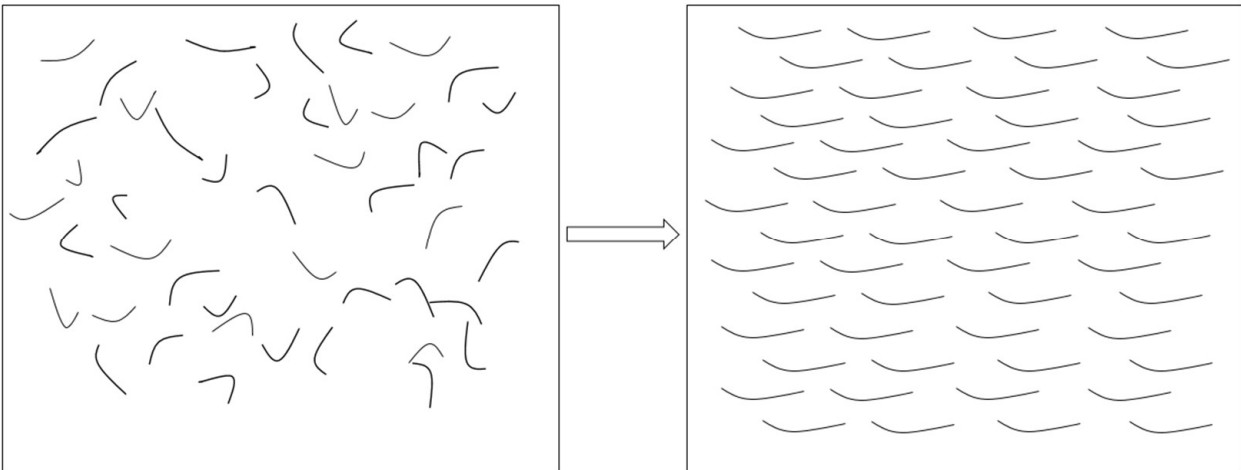

**Figure 8.** Schematic diagram of the effect of dispersion on resistance.

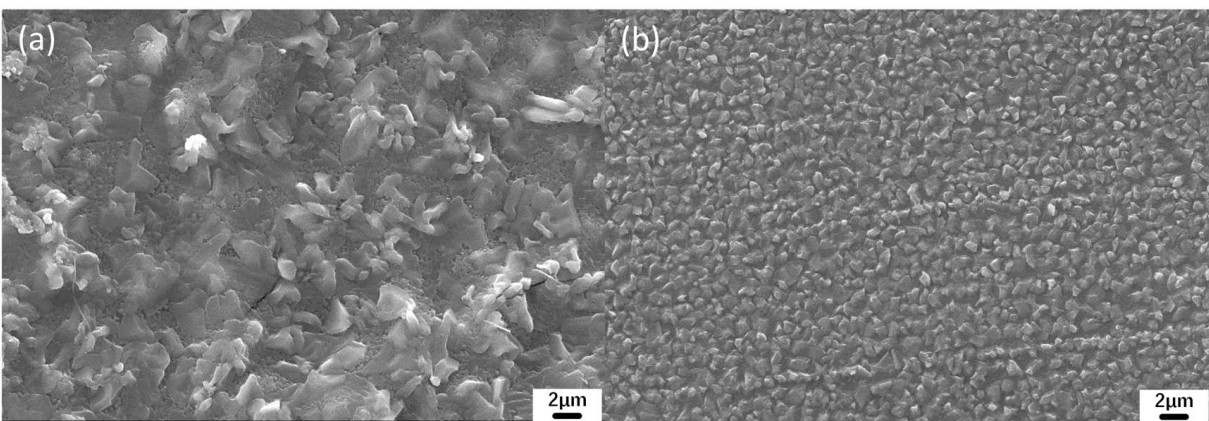

**Figure 9.** SEM images of (**a**) GNs film, (**b**) TiO$_2$/GNs film.

## 4. Conclusions

In this work, TiO$_2$/GNs composite material was successfully synthesized by chemical exfoliation and hydrothermal reduction methods, and its structure was confirmed by XRD, FTIR, and Raman spectra. SEM results show that TiO$_2$ nanoparticles uniformly cover the surface of the GNs. The TiO$_2$/GNs composite was added to sodium silicate to make electrothermal film, and compared with the raw GNs, the heating rate and the maximum temperature were obviously improved. The BET and zeta potential tests prove that the improvement of the performance is due to the enhancement of the specific surface area of TiO$_2$ and the dispersion to the composite by TiO$_2$ particles and sodium silicate. When the dispersion capacity of the GNs is increased, the arrangement between the sheets tends to be more uniform, and more overlap points cause the resistance to decrease. Therefore, TiO$_2$/GNs composite is able to enhance the electrothermal behavior of raw GNs. The new composite material in the field of electric heating has great potential applications.

**Author Contributions:** Conceptualization, C.W.; methodology, W.T.; formal analysis, B.Z.; investigation, W.T. and H.W.; data curation, S.K. and H.W.; writing—original draft preparation, C.W. and B.Z.; writing—review and editing, C.W.; visualization, S.K. and C.Q.; All authors have read and agreed to the published version of the manuscript.

**Funding:** This research was funded by National Science Foundation of China, No. NSFC51302050.

**Institutional Review Board Statement:** Not applicable.

**Informed Consent Statement:** Not applicable.

**Data Availability Statement:** Data is contained within the article.

**Conflicts of Interest:** The authors declare no conflict of interest.

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
