# Peer review of "Hydrothermal Preparation of TiO2/Graphite Nanosheets Composites and Its Effect on Electrothermal Behavior"

_coatings, doi:10.3390/coatings13020226_

Round 1
Reviewer 1 Report (Previous Reviewer 4)
The idea is interesting, however, there are some major points that should be addressed before any possible publications.
1. All of the figures have poor quality, i do suggest authors to use the help of a professional graphical designer.
2. You need to add at least 2 master tables to compare the literature results together.
3. Novelty of this work should be more highlighted in the abstract and introduction.
4. The authors needs to use statistical analysis for all of the numerical results.
5. Some of the references are old (before 2016), authors needs to replace them with new ones.
Author Response
Please see the attachment.

Reviewer 2 Report (Previous Reviewer 2)
I suggest to modify the original title to: “Hydrothermal preparation of TiO2/Graphite nanosheets composites and its effect on electro-thermal behaviors” because you report about 3 different composites.
Please check for lanuage mistakes, i.e. singular – plural
Page 8: Please change the sentence “This result confirms that the white particles in the SEM
photograph are TiO2” to “This result supports the assumption that the white particles in the SEM
photograph are TiO2.”
Page 8: “ It can be seen that the surface area of the TiO2/GNs composite increased to 58.9 m2·g-1
, which is higher than ...” From the absorption plot as given in fig.4 I can not see !
Page 9: “… due to the small amount of absorption water on the .. ” → “… due to the small amount of absorbed water on the .. ”
section 3.2 Electrothermal application of TiO2/GNs composite:
With respect to the intended application of the films the findings reported in this section are of major interest for the reader. Therefore it must be elaborated precisely. Many questions arise when I read the section:
1. sentence“…. heating rate and maximum temperature are .. “, Strictly speaking there is no “maximum temperature”. The correct parameter is the “temperature rise” above the temperature of the surrounding medium. The maximum of the temperature rise is reached when the whole system is in thermal equilibrium. The “maximum temperature” in air of 20°C is completely different from the “maximum temperature” in water of 50°C. Therefore it is necessary to define the ambient medium and the starting temperature of the heating experiments. The presence of a glass substrate even complicates the situation.
In fig. 7. Do you start to heat up from 0°C. As it could be assumed from the fig. or was the initial temperature different. Then it must be mentioned in the caption.
Are the samples as described above? 150 mm × 75 mm ×0.08 mm film on a 5 mm thick glass plate?
How the contacts are made for (1) heating experiment (2) four probe resistivity measurements ?
As the authors suspect the electrical conductivity could be anisotropic. (Expressed by fig 8??)
What actions were taken to support or reject this assumption?
The only information given for the heating experiment was a voltage of 15V. DC or AC ? What current and what electrical power resulted from the applied voltage?
In fig. 1 it will ease the readers understanding when the different peaks are labeled as defined in the text (Eg(1), B1g, ….).
From fig 2 - 7 it remains unclear what type of TiO2/GN composite (100°C, 120°C or 140°C) was under examination. The same applies to the text.
In fig. 2c the red dot is difficult to detect. It can be helpful to place the label “Spot 3” near by.
Fig 2d is of poor resolution. Magnifying the EDX plot does not lead to a readable resolution.
In fig.5. are the same arbitrary units and vertical axis origin used for the two plots shown or are they separated for clarity. In the second case it should be noted in the caption. (In other words is GNS more transparent than TiO2/GNS.)
Author Response
Please see the attachment.

Reviewer 3 Report (Previous Reviewer 1)
This revised paper has been reviewed again. The authors revised the paper according the reviewer's comments. The reviewer considers that this version is acceptable.
Author Response
Thank you so much for your comments!
Besides, we want to add Kang Sibo as the Corresponding author. Because he has done lots of later work and given plenty of advice.
Round 2
Reviewer 2 Report (Previous Reviewer 2)
This version now is clearly improved. I have only 3 suggestions for improvement:
Table 2. Although the term “Kratio” is sometimes used my preference is “K-Ratio”.
In table 3 In the column “sample” for reference 24 it appears to consist of two separate samples due to the formatting.
Currently the transition from section 4 “Conclusions” to the references is abrupt and should be reformatted to include a section 5 “References”
Round 3
Reviewer 2 Report (Previous Reviewer 2)
The manuscript is revised fine.
Author Response
Thanks so much for your comments!
This manuscript is a resubmission of an earlier submission. The following is a list of the peer review reports and author responses from that submission.
Round 1
Reviewer 1 Report
The authors of the manuscript entitled "Hydrothermal preparation of TiO2/Gr nano-sheet composite and its effect on electrothermal behaviors" present the synthesis of heat-generating materials based on TiO2-modified graphite.
The relevance of the proposed research is in doubt. Many misleading sentences are in the manuscript, e.g., the authors use the terms "graphite and graphene" to identify the same material. The detailed comments are shown below. The reviewer cannot recommend this manuscript for publication in Coatings.
(1) The authors give the vague definition of scientific novelty.
(2) In the Abstract, the authors write that they prepared the TiO2/graphite nano-sheet composite; however, in the Introduction, the authors describe the advantages of graphene and its modification strategies. But the authors didn't reveal the reason for selecting the graphite materials for their research. Therefore, the Introduction reviews the literature, not corresponding to the research topic.
(3) The last paragraph of the Introduction is confusing; the authors argue "Hence, we modified graphene by TiO2 to study the synergistic electrothermal effects in the properties enhancement of sodium silicate composites film. It is anticipated that loading TiO2 nano-particles on the graphene nano-sheets could prevent the graphene nano-sheets from restacking and reduce the resistivity compared with directly adding TiO2." and then they claim "This work attempts to provide a promising electrothermal material so as to enhance the graphite nano-sheets electrothermal behavior." Were the authors unable to determine the type of obtained material? Graphene or graphite?
(4) Section 2.2. Preparation of graphite nano-sheet/TiO2 hybrid: "The P25–grapheme was obtained via a hydrothermal method based on Qiu’s article (reference No. [12]) with modifications." This is misleading because the P25 is commercial TiO2 powder, containing the mixture of anatase and rutile phases while the authors used the titanium oxysulfate (TiOSO4) to obtain the TiO2 particles with the anatase structure.
(5) Characterization methods (section 2.3) were not fully described; it is critical to include sufficient information about the experimental conditions, e.g., the wavelength of laser excitation for Raman spectroscopy, a model of zeta potential calculation and others.
(6) Section 3.1. Characterization of the TiO2/GNS composite: "Raman spectra of TiO2 has strong Raman characteristic peaks at 152 cm-1(Eg(1)), 396 cm-1(B1g), 516cm-1(A1g), and 639cm-1(Eg(2)), respectively [14]." The authors could specify that these Raman modes are attributed to a pure anatase phase of TiO2.
(7) Section 3.1. Characterization of the TiO2/GNS composite: "After the TiO2 nanoparticles loaded on graphene, the peaks of TiO2 appeared on the Raman spectra of all the temperature samples, which reveal the successful fabrication of graphene/TiO2 hybrid." This sentence is misleading because the graphene can be identified by the peak intensity ratio of the 2D and G bands (I2D/IG). The obtained Raman spectra clearly demonstrate a sharp G band and an asymmetric 2D band, which refers to the graphite. The graphite structure is also confirmed by the obtained XRD results.
(8) Section 3.1. Characterization of the TiO2/GNS composite: "Furthermore, the G band of graphene/TiO2 shows a shift to higher wave numbers and 2D band to lower wave numbers compared to that of bare graphene, which is attributed to the effect of doping and the reduced number of sheets, indicating that the re-stack of graphene sheets was controlled." The explanation of shifts should be revised in the context of graphite changes, strains or synthesis conditions.
(9) Section 3.1. Characterization of the TiO2/GNS composite: "The film consists of a number of particles less than 5nm, and we speculate that those large particles may be made up of small particles due to the concentration of the reaction at the defect." How was the reported value determined? The SEM images don't confirm the presence of primary particles and the nanoscale nature of TiO2 particles. The characterization of the primary particles by SEM requires greater magnification and higher resolution. Moreover, Fig. 2(c) caption refers to Fig. 2(b) not (a) and the "red point", corresponding to Fig. 2(d) is absent in Fig. 2(c).
(10) Section 3.1. Characterization of the TiO2/GNS composite: "However, it is still lower than theoretical specific surface area (2630 m2 · g-1) maybe due to the incomplete exfoliation of graphite oxide and the agglomerations of graphene layers during reduction process[18][19]." The authors make groundless arguments, not correlating with the result data.
(11) The reference [18] duplicates the reference [19].
(12) Section 3.1. Characterization of the TiO2/GNS composite: "It can be seen that after the thermal reduction of the graphite nano-sheets, the oxidation groups on the surface have been completely removed, only a little carbon-hydrogen bonds can be detected." Сompared to what? Please clarify.
(13) Section 3.1. Characterization of the TiO2/GNS composite: "The zeta potential of TiO2/GNS has an absolute value of about 30, higher than that of GNS about 15. Absolute value of zeta potential is an important indicator of dispersion stability, the larger number means the better the dispersion." The magnitude of the zeta potential up to 30 is classified as the incipient instability. Is this magnitude enough to stabilize the particles or resist the agglomeration?
(14) The reviewer has a question about the electrothermal properties. Are the electrothermal parameters of the obtained TiO2/GNS film, such as the heating rate, the maximum temperature and the conductivity higher or comparable to those of developing analogues at the same electric power? Please give examples.
(15) The Conclusion should be revised in accordance with the above-mentioned comments.
(16) The authors use the references published before 2015. The reviewer can conclude that the research area hasn't developed since 2015, or the authors don't have the relevant information in the research field.
(17) The manuscript doesn't include the description of Fig. 8 and doesn't refer to it.
Reviewer 2 Report
“Hydrothermal preparation of TiO2/Gr nano-sheet composite and its effect on electrothermal behaviors”
Please complete the title: “Hydrothermal preparation of TiO2/Graphite(?) nano-sheet composite and its effect on electrothermal behaviors”
The manuscript suffers from poor English which makes it difficult to distinguish between language mistakes and scientific errors. The following can illustrate this uncertainty : do the authors mean graphite with a 3 dimensional structure or graphene with a two dimensional structure composed of a flat monolayer of carbon atoms.
It is an interesting idea to add TiO2 to graphene(!) in order to enhance the electrical sheet conductance. However the introduction does not give an explanation of the necessity nor does it give an overview of the current state of the art. In many applications it is common practice to make the conductive sheets thicker in order to increase the sheet conductance. In the introduction the authors do not even mention the existence of (transparent) conducting oxides (TCO) such as Indium-Tin-Oxide which are currently a popular choice for thin heating films.
The introduction does not explain what are the advantages to shift towards carbon based heating foils. Especially when they will be incorporated into wearables. In this case however the mechanical properties of the films are of primary interest.
Ref [3] refers to Carbon Nanotubes not to Graphite and/or graphene.
Ref [6] does not support the statement given by the authors: “Despite the aforementioned merits, graphene can’t satisfy all demands for many different applications [6]”
Ref [19] equals Ref [18]
Table 1 does not mention any Graphene or TiO2
In Fig.2 If 2(a) is mere GNS and 2(b) is TiO2/GNS I wonder what 2(c) is showing? If it is an enlargement of 2(a) as written in the caption there should not be any Ti shown in the spectra of 2(d)
furthermore I can not see any “red point” indicating the location from where the EDX is taken.
From fig.4 I derive that the TiO2/GNS layer is (i) either thicker than the GNS sheet or (ii) it contains more voids in the same volume. In the text you mention that TiO2/GNS layers have a resistivity which is only ½ of the one for the mere GNS layer. For a more porous material however a higher resistivity is more likely.
For very thin layers it is favorable to determine the (2 dimensional) sheet resistivity or sheet conductivity. The resistivity is only meaningful for 3 dimensional conductors.
What is shown in fig.8 ?
Reviewer 3 Report
Unfortunately, I could not recommend presented manuscript entitled “Hydrothermal preparation of TiO2/Gr nano-sheet composite and its effect on electrothermal behaviors” for publication. Despite interesting topic, the whole presentation of article is too poor and needs to be revised.
· First of all, grammar and sentences’ construction need to be corrected. Some of the sentences do not have any sense, lot of them look like directly took from online translator.
· Secondly, there is a lack of consequence. Authors ones write “graphite” ones “graphene”. Ones TiO2 is prepared from TiOSO4 precursor, ones is described as “P25”. This looks like Authors do not know what they were doing.
· Only due to one sentence: “We can see the 140 °C sample has the highest peak of TiO2, it seems that the temperature is more suitable” and Fig. 1, I realized that at the beginning Authors tried to obtain TiO2/GNS samples in different hydrothermal process temperatures, but later they gave up on this (or they have chosen the “best” sample only on the base of Raman spectra).
· Description of novelty, results discussion and conclusions need to be expanded.
· I also do not understand if Authors analyzed TiO2/GNS powder or “glass plate films”
· Some superscripts and subscripts are missing.
· In Authors’ contribution: “Wang Chunyu and Zhong Bo wrote the main manuscript text and Qin Chunlin prepared figures”. In this regard, who conducted the research? What other Authors were doing?
· Before submission to MDPI journals, manuscript must be put in a special form.
· 18 publications and 1 patent – this number is quite low for research paper. Moreover, the latest publication was from year 2015.
Concluding, I stand by my decision about rejecting of presented manuscript.
Reviewer 4 Report
The idea is interesting; however, extensive revision is required before possible publication.
1. Introduction part should be expanded to cover most of the recent studies in this field, and distinguish the previous novelties and lack of innovations.
2. For the surface morphology characterizations, AFM and TEM should be added as well.
3. To investigate the hydrodynamic particle size, DLS should be added.
4. All of the numerical results should be investigated by the statistical analysis.
5. At least 2 master tables should be added to compare the results with the literature.
6. Some of the references are old, remove them (before 2016) and replace them with the new ones.
7. What is the reason for broadenings the XRD?
8. Mapping should be added to the SEM images.
Major revision is required